# Genetic differentiation at extreme latitudes in the socially plastic sweat bee *Halictus rubicundus*

Bas A. Michels[1], Mariska M. Beekman[1], Jeremy Field[2], Jodie Gruber[2], Bart A. Pannebakker[1], Charlotte Savill[3], Rebecca A. Boulton[1,4]*

1 Laboratory of Genetics, Plant Sciences Group, Wageningen University and Research, Wageningen, Netherlands, 2 College of Life and Environmental Sciences, University of Exeter, Penryn Campus, Penryn, United Kingdom, 3 Faculty of Biology, School of Health Science, Medicine and Health, University of Manchester, Manchester, United Kingdom, 4 Biological and Environmental Sciences, University of Stirling, Stirling, United Kingdom

* rebecca.boulton@stir.ac.uk

**Data Availability Statement:** Data and code are available at osf.io/s8txc.

**Funding:** This work is part of a project that received funding from the European Research

## Abstract

The sweat bee *Halictus rubicundus* is an important pollinator with a large latitudinal range and many potential barriers to gene flow. Alongside typical physical barriers, including mountain ranges and oceans, the climate may also impose restrictions on gene flow in this species. The climate influences voltinism and sociality in *H. rubicundus*, which is bivoltine and can nest socially at warmer lower latitudes but tends to be univoltine and solitary in the cooler north. Variation in voltinism could result in phenological differences, potentially limiting gene flow, but a previous study found no evidence for this in *H. rubicundus* populations in mainland Britain. Here we extend the previous study to consider populations of *H. rubicundus* at extreme northern and southern latitudes in the UK. We found that bees from a population in the far north of Scotland were genetically differentiated from bees collected in Cornwall in the south-west of England. In contrast, bees collected across the Irish Sea in Northern Ireland showed slight genetic overlap with both the Scottish and Cornish bees. Our results suggest that when populations at extreme latitudes are considered, phenology and the climate may act alongside physical barriers such as the Scottish Highlands and the Irish Sea to restrict gene flow in *H. rubicundus*. We discuss the implications of our results for local adaptation in the face of rapidly changing selection pressures which are likely under climate change.

## Introduction

The sweat bee *Halictus rubicundus* (Hymenoptera: Halictidae) has a broad geographic range, is abundant in many regions and can be found nesting in large aggregations, often of consisting of several hundred bees. This Holarctic ground-nesting bee has been well studied throughout mainland Europe, the UK and North America. *Halictus rubicundus* forages on a wide range of plant species (it is polylectic) and so it is likely a key pollinator for many native plant

Council (ERC) under the European Horizon's 202 research and innovation programme (grantagreement no. 695744). RAB was funded by a Wageningen Graduate School Postdoctoral Talent fellowship and a BBSRC discovery fellowship.

**Competing interests:** The authors have declared that no competing interests exist.

species, particularly at high latitudes where pollinator diversity is lower [1,2]. Voltinism and social structure have been well-studied in *H. rubicundus* and have been shown to vary across latitudes and altitudes [3–5]). Variation in social structure in *H. rubicundus* is thought to be related to the climate, as cooler temperatures and a shorter growing season restrict the number of foraging days. This limits bees in colder climates to univoltinism which precludes sociality (Fig 1; see also [3]. In the south of its geographical range, bivoltinism and a second, social, 'worker' generation are typical while in the north (and at higher altitudes), *H. rubicundus* populations tend to be univoltine with one solitary generation per year [4–6]. This is the pattern we see in North America and the UK; populations are solitary in cooler climates and social where it is warmer [4–7]. Additionally in the UK, sociality has been shown to be plastic; when solitary northern bees are transplanted to the south, they express bivoltinism and sociality [4,6,7].

Research on bees and other insects suggests that regular spells of very hot weather caused by anthropogenic climate change are contributing to declines in species diversity and abundance worldwide [8,9]. However, Schürch et al. [10] suggested that warming temperatures due to climate change may actually benefit *H. rubicundus* numbers in the UK in the short term. By combining foraging data with climate forecasts, they hypothesised that northern populations would transition to social nesting under most climate change scenarios, increasing the numbers of provisioning bees in the summer which may actually alleviate the pollinator crisis [11]. Whether or not this benefit of climate change will be realised will depend on a number of factors. For instance, local adaptation to the climate, synchronicity with regional food plant phenology and barriers to gene flow can all influence the evolutionary trajectory and persistence of *Halictus rubicundus* under a changing climate.

Gene flow between populations with different phenologies and social structures is particularly salient with respect to climate change. Variation in social organisation has been proposed to restrict gene flow in *H. rubicundus*, perhaps due to non-matching phenologies of social versus solitary populations [5]. This scenario may arise if, under solitary nesting, the first generation of reproductives (B1) does not emerge at the same time as the reproductive generation (B2) produced by social nests (Fig 1). This might limit mating opportunities between social

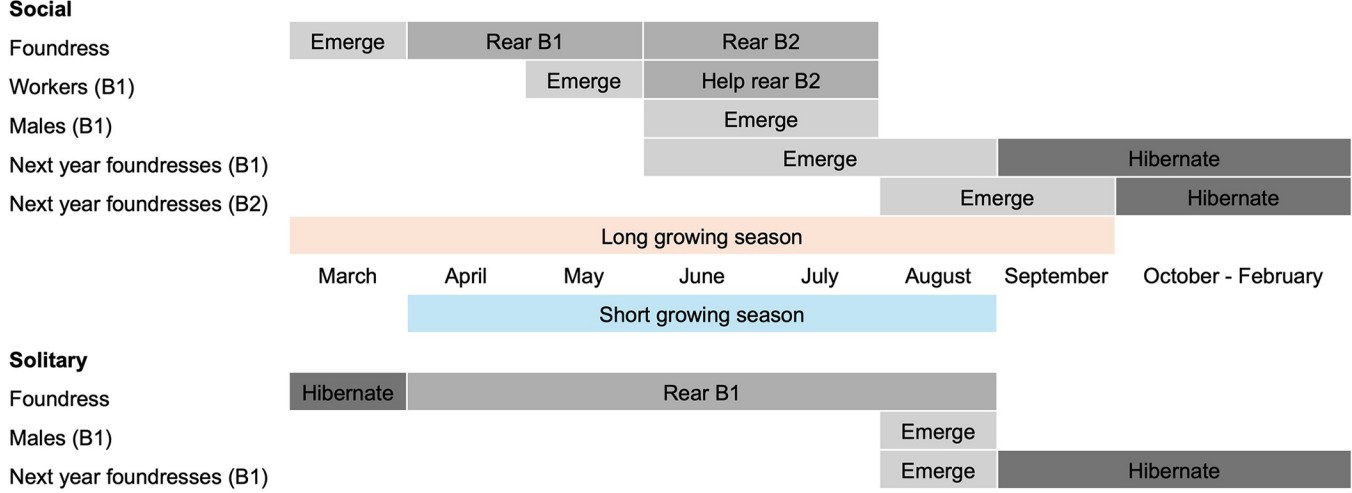

**Fig 1. Life cycle of *Halictus rubicundus*.** In solitary populations, the foundress rears one brood of both sexes (B1), which subsequently mate. After mating, the B1 males die and the B1 females hibernate underground and become foundresses the next year. In social populations, the foundress also rears a mixed B1 brood. Of this B1 brood, some females become foundresses themselves and rear a brood of their own in the same year, while others go into hibernation until the following year (not shown in figure). The other females of the B1 brood become workers; assisting the original foundress in provisioning and raising a second brood (B2) of both sexes (figure adapted from [10]).

and solitary populations, particularly if males are short-lived (but see [12]). When this results in reduced or no exchange of genetic material between the two phenotypes, a long-term phenological barrier to mating may arise, resulting in larger genetic differences between populations over time. Soucy and Danforth [5] found evidence for this in North American populations of *H. rubicundus*, but in the UK there appears to be substantial gene flow between solitary and social populations in the north and south respectively [4]. Indeed, Soro et al. [4] found evidence that only the Irish Sea poses a barrier to gene flow in British *H. rubicundus*. However, as yet, no studies have sampled populations from the full latitudinal range of *H. rubicundus* in the UK or Europe which limits our ability to predict how climate change will impact this species and the ecosystems that may depend on it.

In this study, we investigate whether the previous findings, the Irish Sea as the only barrier to gene flow, extend to populations at the latitudinal extremes of the range of *H. rubicundus* in the UK, or whether other barriers might also limit gene flow in this species. For instance, the Scottish Highlands are another potential geographic barrier which could limit geneflow, but non-physical barriers could also play a role. Males and females from populations with different social phenotypes may be less likely to encounter each other and mate, for example. Local adaptation to different microclimates and food plants/phenologies could also reduce the fitness of hybrids, limiting effective gene flow between populations that have experienced different selective environments. We sampled bees from three previously unstudied populations of *H. rubicundus*, one in the far north (Scotland, N 57˚) and two in the south-west (Cornwall, England, N 50˚) of its range. Using twelve microsatellite markers, we assessed patterns of genetic variation within and between these three populations and compared this to genetic variation observed in the Belfast population previously studied by Soro et al. [4]. These results help improve our understanding of barriers to gene flow in *H. rubicundus* and other socially polymorphic species and can help us better predict the effects of climate change on genetic diversity and population persistence in important pollinators such as *H. rubicundus*.

## Materials and methods

### Sample collection

We sampled *Halictus rubicundus* females (N = 142) from four aggregations across the UK (see Fig 2). In Belfast (at the same site studied by Soro et al. 2010 [4]) we collected 39 females by excavating them during hibernation in February 2020. In Migdale (Scotland; N = 40) and Boscastle (N = 28) (Cornwall) foundresses were collected by hand netting in May and June 2018. In Bodmin (Cornwall, N = 35) bees were collected (again by hand-netting) in May and June 2019. All specimens were placed in 99% ethanol and stored at -20˚C until DNA extractions were performed.

### DNA extractions

DNA was extracted from all ethanol-stored individuals using a hot-shot extraction method [13]. A hind leg was removed from each individual and left to dry on absorbent paper, after which it was cut into small pieces using dissecting scissors and placed into a single well of a 96-well plate. Fifty μL alkaline lysis buffer was added to each sample which were subsequently covered and placed in a PCR machine at 95˚C for 45 minutes after which a neutralizing reagent (40mM Tric-HCl in water at pH 5) was added. The plate was stored at -20˚C until the extractions were used in multiplex PCR reactions detailed below.

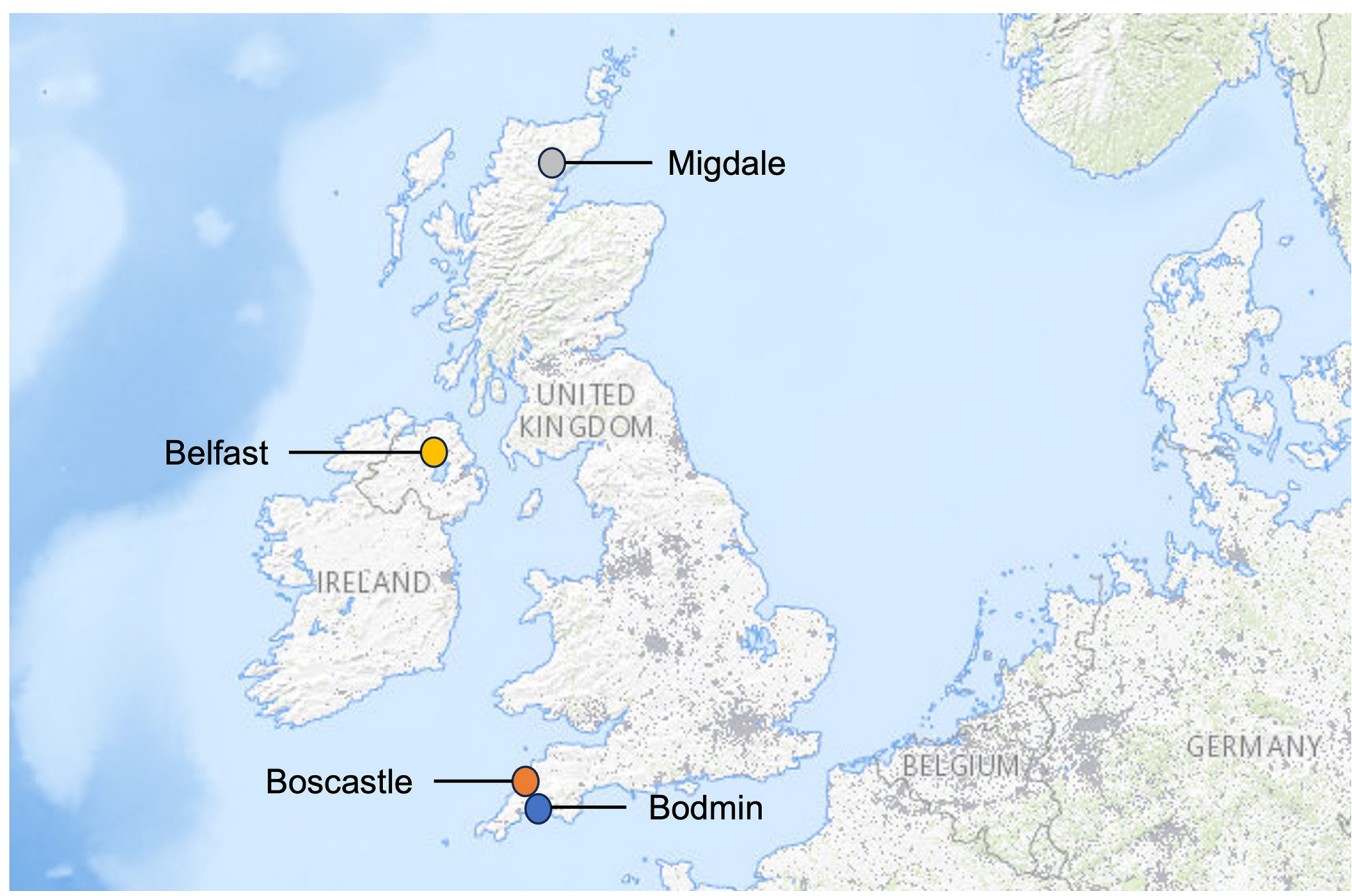

**Fig 2. The sampling locations of the *Halictus rubicundus* aggregations used in this study.** This image was reproduced from [7] under the terms of the Creative Commons Attribution licence (CC BY).

## Microsatellite amplification

We used 13 microsatellite markers that were previously designed for *H. rubicundus* by Soro and Paxton [14] to assess genetic variation within and between the four sampled populations. Of these 13, only 12 amplified successfully for all samples (rub61 did not, see S1 Table). We genotyped each individual at the 12 remaining loci. We ran three multiplex reactions using fluorescently labelled primers (see S1 Table and Soro and Paxton 2009 for details; set 1: rub02, rub30, rub35, rub37b, rub59; set 2: rub04, rub06, rub55, rub60; set 3: rub73, rub80, rub72). PCR reactions were carried out in 5 μL reactions and contained: 1 μL DNA template, 4 μL PCR mastermix (made up of 0.75 μl at a concentration of 0.2 μM of each primer, 80 μL ultra-pure water, and 250 μL QIAGEN Multiplex PCR mix; QIAGEN Inc. Cat. No. 20614). PCR conditions were as follows: initial denaturation at 95˚C for 5 min, followed by 35 cycles of 95˚C for 30 s, 60˚C for 90 s, 72˚C for 90 s, and final extension at 68˚C for 30 min. PCR amplification was performed using a DNA Engine Tetrad ® Thermal Cycler (MJ Research, Bio-Rad, Hemel Hempstead, Herts, UK). Fragment separation was performed on an ABI 3730 48-well capillary DNA Analyser using the LIZ size standard (Applied Biosystems Inc, Waltham, USA. Cat. No. 4322682). Allele sizes were scored by a single scorer (BM) using Geneious v9.1.5 (Biomatters Ltd, Auckland, New Zealand). Peaks were screened for quality, based on the recommendations of [15]. Any peaks that did not meet quality standards were excluded from

analysis of that individual, and if one individual had multiple exclusions, it was excluded entirely.

We excluded scores from one locus (rub04) from the analysis due to a high frequency of null alleles (~20% in all populations; results based on analysis using FreeNA; Chapuis and Estoup 2007) and significant deviation from Hardy-Weinberg equilibrium (HWE) for this locus across all populations. Population genetic analyses were performed using GenAlEx 6.5 [16,17]. We tested whether all markers conformed to HWE and calculated population descriptive statistics including observed ($H_o$) and expected heterozygosity ($H_e$), $F_{ST}$ and Nei's genetic distance. GenAlEx was also used to conduct a principal component analysis (PCoA) on the genetic distances to visualise genetic differentiation within and between populations. The total number of alleles ($N_a$) and private alleles ($N_p$; alleles unique to each population) per marker per population (rarefied to a common sample size of 28) were calculated in *ADZE* [18] for $N_p$, and with R v. 4.2.1 [19] in RStudio v. 2022.07.1 [20] using the package *pegas* v. 1.1 [21] for $N_a$.

## Ethical approvement statement

This research project complies with ethical legislation laid out by the EU, the UK (Animals Scientific Procedures Act 1986) and the University of Exeter and WUR. No human subjects, animals covered by the Animals (Scientific Procedures) Act or genetically modified organisms were involved in this research. This work adhered to the ethical guidelines set out by ASAB and the 3Rs to ensure that results are reliable and replicable.

## Results

The mean expected heterozygosity ($H_e$) across all four populations was 0.79 and the observed heterozygosity *($H_o$)* was 0.74. There were slight differences in observed ($H_o$) and expected ($H_e$) heterozygosity across populations (Table 1). For all populations, $H_o$ was lower than $H_e$ and this was particularly pronounced in Bodmin.

The average number of alleles across all loci was 10.9. Boscastle had the highest number of alleles across all loci ($N_a$ = 12.27, Table 2) and Migdale had the lowest ($N_a$ = 8.52). The population with the most private alleles ($N_p$) was Belfast ($N_p$ = 2.74). Migdale had the fewest private alleles ($N_p$ = 1.08). The marker with the lowest number of alleles was rub37b ($N_a$ = 5), rub30 had the most alleles across all populations ($N_a$ = 39) and the most private alleles within each population ($N_p$ = 3.6 on average).

Significant deviation from HWE was observed at several loci (see Table 3). Marker rub73 deviated significantly from HWE in all populations. The Bodmin population showed significant deviation from HWE at 8/11 loci.

Pairwise $F_{ST}$ values and Nei's genetic distances showed the most genetic similarity between Bodmin and Boscastle ($F_{ST}$ = 0.023; Nei's = 0.241, Table 4). Migdale was the most genetically different from all other populations (Bodmin: $F_{ST}$ = 0.061; Nei's = 0.618. Boscastle: $F_{ST}$ = 0.059; Nei's = 0.625. Belfast: $F_{ST}$ = 0.054; Nei's = 0.491). Bees collected in the south of mainland

**Table 1. Observed ($H_o$) and expected ($H_e$) heterozygosity ± standard error across four aggregations of *H. rubicundus* in the UK.**

|  | $H_o$ ± SE | $He$ ± SE |
|---|---|---|
| Bodmin | 0.63 ± 0.03 | 0.81 ± 0.02 |
| Boscastle | 0.81 ± 0.04 | 0.83 ± 0.02 |
| Migdale | 0.73 ± 0.04 | 0.75 ± 0.03 |
| Belfast | 0.78 ± 0.04 | 0.80 ± 0.04 |

**Table 2. Population genetic statistics for *Halictus rubicundus* from four populations across the UK.**

| | | Population average | rub37b | rub02 | rub59 | rub30 | rub35 | rub60 | rub06 | rub55 | rub72 | rub80 | rub73 |
|---|---|---|---|---|---|---|---|---|---|---|---|---|---|
| Number of alleles ($N_a$) | Bodmin | 11.34 | 5.00 | 7.76 | 17.79 | 15.64 | 18.32 | 11.47 | 7.56 | 7.75 | 12.78 | 11.32 | 9.39 |
| | Boscastle | 12.27 | 5.00 | 10.00 | 14.00 | 18.00 | 19.00 | 13.00 | 8.00 | 7.00 | 14.00 | 16.00 | 11.00 |
| | Migdale | 8.52 | 2.91 | 7.61 | 12.15 | 15.16 | 11.06 | 12.30 | 5.70 | 7.10 | 7.52 | 6.58 | 5.61 |
| | Belfast | 11.50 | 4.85 | 7.96 | 12.81 | 24.02 | 17.93 | 5.34 | 10.54 | 9.74 | 14.81 | 9.92 | 8.56 |
| Mean number of alleles per marker | | | 4.44 | 8.33 | 14.19 | 18.20 | 16.58 | 10.53 | 7.95 | 7.90 | 12.28 | 10.96 | 8.64 |
| Number of private alleles ($N_p$) | Bodmin | 1.65 | 0.00 | 1.00 | 5.68 | 1.57 | 0.36 | 1.26 | 1.80 | 0.99 | 3.68 | 0.86 | 0.99 |
| | Boscastle | 1.91 | 0.00 | 1.00 | 2.20 | 1.77 | 4.09 | 1.82 | 0.11 | 0.00 | 2.50 | 4.29 | 3.21 |
| | Migdale | 1.08 | 0.00 | 1.00 | 0.70 | 3.30 | 0.28 | 2.13 | 0.70 | 2.70 | 0.00 | 0.99 | 0.13 |
| | Belfast | 2.74 | 0.00 | 0.00 | 1.20 | 7.76 | 5.51 | 0.00 | 5.62 | 2.92 | 3.56 | 2.62 | 1.00 |
| Mean number of private alleles per marker | | | 0.00 | 0.75 | 2.44 | 3.60 | 2.56 | 1.30 | 2.06 | 1.65 | 2.43 | 2.19 | 1.33 |

Number of alleles ($N_a$) per population and/or marker, along with the rarefied number of alleles ($N_p$) per population and/or marker.

Britain (Bodmin and Boscastle) were more genetically similar to bees across the Irish Sea (Belfast) than they were to bees collected in the North of mainland Britain in Migdale (Bodmin: $F_{ST}$ = 0.046; Nei's = 0.504. Boscastle: $F_{ST}$ = 0.044; Nei's = 0.503). Principal component analysis (PCoA) confirmed this visually (Fig 3). The first three principal components explain 7.64%, 5.74% and 4.61% of the variance observed. The Bodmin and Boscastle samples show considerable overlap. There is a slight overlap between Belfast and the other populations, but Migdale samples form a distinct cluster (Fig 3).

## Discussion

Barriers to gene flow in the ground-nesting sweat bee *Halictus rubicundus* have been suggested to be both geographic and temporal in nature [4,5]. Geographic barriers such as the Rocky Mountains and the Irish Sea appear to restrict gene flow across this species Holarctic range, but the existence of a temporal or phenological barrier to gene flow is more ambiguous. In North America, it appears that geographically close populations with different social structures are genetically isolated from one another [5]. This may be due to temporal differences in the emergence of reproductives which limit mating between social and solitary populations. In the UK, where the social organisation of *H. rubicundus* is plastic, no evidence for temporal barriers restricting gene flow have been found [4,6].

In this study, we extended the work of Soro et al. [4] to consider British populations of *H. rubicundus* at extreme latitudes. We found the greatest genetic difference between bees

**Table 3. Departure from Hardy-Weinberg equilibrium across microsatellite markers in each of the four *Halictus rubicundus* population assayed.**

| Set | 1 | | | | | 2 | | | 3 | | | |
|---|---|---|---|---|---|---|---|---|---|---|---|---|
| | rub37b | rub02 | rub59 | rub30 | rub35 | rub60 | rub06 | rub55 | rub72 | rub80 | rub73 | |
| Bodmin | ns | *** | ** | *** | * | *** | * | ns | ns | *** | * | |
| Boscastle | ns | ns | ns | ns | ns | ns | *** | ns | ns | ns | * | |
| Migdale | ns | ns | ns | *** | ns | ns | ns | ns | ns | *** | *** | |
| Belfast | ns | ns | ns | * | ns | ns | ns | ns | ns | *** | *** | |

$P < 0.05$ =

*, $P < 0.01$ =

**, $P < 0.001$ =

***, ns = not significant.

**Table 4. The $F_{ST}$ and Nei's genetic distances between the *Halictus rubicundus* populations assayed.**

|  | Bod | Bos | Mig | Bel |
|---|---|---|---|---|
| Bod | 0.000 | 0.241 | 0.618 | 0.504 |
| Bos | 0.023 | 0.000 | 0.625 | 0.503 |
| Mig | 0.061 | 0.059 | 0.000 | 0.491 |
| Bel | 0.046 | 0.044 | 0.054 | 0.000 |

On the bottom left side (dark grey) Nei's genetic distance is displayed, while on the top right (white) the $F_{ST}$ values are shown.

collected in mainland UK; between those collected in Migdale, in the far north of Scotland, and those from South-West England (Cornwall). In contrast, bees collected across the Irish Sea in Northern Ireland showed slight genetic overlap with both Scottish and Cornish bees.

While our results do not allow us to comment on the exact nature of the barriers to gene flow, they do suggest that other barriers exist. Moreover, one important barrier, the Irish Sea, highlighted by the study of Soro et al. [4], did not appear to restrict gene flow to as great an extent as any potential barriers between the north of Scotland and the south of the UK. Many barriers could be responsible for the genetic isolation of the Scottish population, including the Scottish Highlands, local adaptation (e.g. climate) or consistent differences in phenology (e.g., due to the climate or social organisation). Longitudinal studies which sample from more populations with known phenologies (and social phenotypes) are needed to elucidate the most likely mechanisms.

We found that many markers deviated from HWE, particularly for bees sampled in Bodmin. This brings the caveat that many of the results we present should be interpreted with a degree of caution. Future studies using whole genome sequencing and genotyping-by-sequencing will provide greater power to assess patterns of genetic variation within, and genetic differences between, populations. Sequencing the same individuals that we used here would also allow us to validate the results of this microsatellite-based study [22].

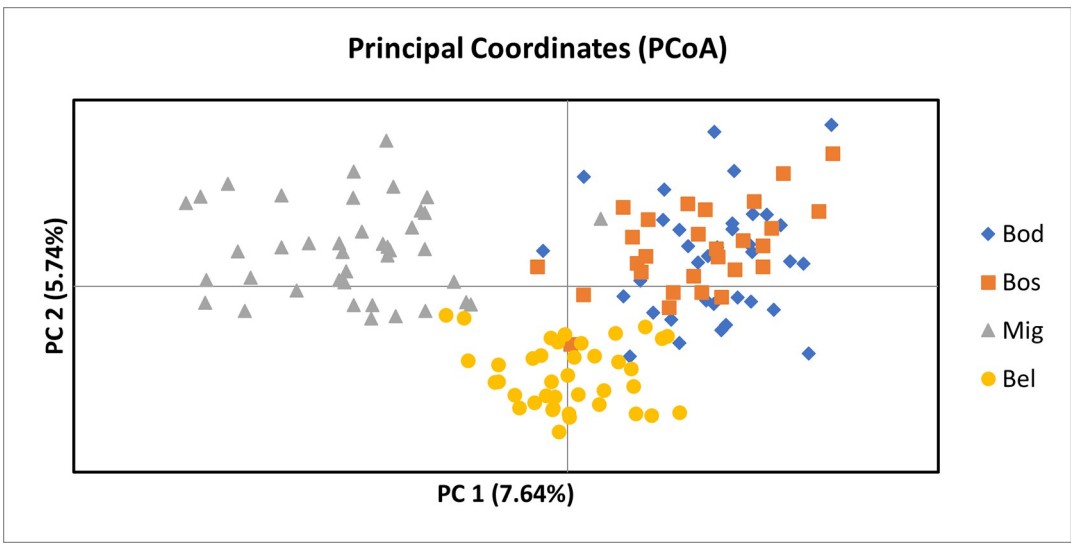

**Fig 3. Principal component analysis (PCoA) of multilocus genotypes from four populations of *Halictus rubicundus* from the UK.** Cornwall (south-west UK); Bodmin (Bod): Blue diamonds, Boscastle (Bos): Orange squares. Belfast (Northern Ireland; Bel): Yellow circles. Migdale (Scotland; Mig): Grey triangles.

In addition to our findings regarding genetic differences between populations, we also found low observed compared to expected heterozygosity in all populations, particularly Bodmin, suggesting that founder effects may have reduced genetic variation. The founder effect is problematic for small populations where genetic drift and inbreeding lead to a random loss of alleles, and thus a decrease in heterozygosity. The Bodmin population showed the largest deficit in observed heterozygosity compared to expected heterozygosity and also showed deviation from HWE in 8/11 markers (for all other populations, this was only two or three out of 11). These results suggest that genetic variation has been reduced in Bodmin, but interestingly, not in the nearby population at Boscastle. Bodmin is at a higher altitude than Boscastle, the climate is cooler, wetter, and less sheltered, and food is scarcer. We have noticed that aggregations from Bodmin tend to be smaller and more transient than other populations we monitored every spring and summer between 2018 and 2021.

The founder effect and isolation may have resulted in low observed heterozygosity in Bodmin (which could also contribute to the deviations from HWE that we observed) and this may have led to inbreeding and reduced population fitness. For Hymenoptera, the most direct effect of inbreeding on population fitness arises through sex determination. Hymenoptera are haplodiploid; diploid females are produced from fertilised eggs and haploid males from unfertilised eggs. Under inbreeding diploid males are sometimes produced from fertilised eggs if the gamete is homozygous at the sex-determining loci. In many Hymenoptera there is only one sex-determining locus, this is known as single locus complementary sex determination (sl-CSD [23]) and so the likelihood of diploid male production is high when inbreeding is common. Diploid male production imposes a genetic load on the population because diploid males are effectively sterile, potentially resulting in a 'diploid male vortex of extinction' [24]. While the mechanism of sex determination in *H. rubicundus* is unknown, the presence of high levels of male diploidy in *Halictus poeyi* in Florida [25] suggests the diploid male vortex could be a problem for *H. rubicundus* as well. The costs of inbreeding may have acted alongside environmental stressors (such as poor weather and high levels of cuckoo parasitism) at Bodmin resulting in the reduced nesting activity that we observed between 2018 and 2020.

The Migdale population was the most stable in terms of numbers over the several years that we have been monitoring these aggregations with several thousand nests found on a 20 m-long stretch of bank. Our analyses show that despite its size, the Migdale population had the lowest number of alleles per marker on average; 8.5 compared to around 11–12 for the other three populations. The fact that Migdale also had the lowest $H_e$ value reflects this low allelic diversity. Unlike the previous study by Soro et al. [4], who found that the Belfast population across the Irish Sea showed the most genetic differentiation from populations in mainland UK, we found that another mainland UK population (Migdale in Scotland) was the most genetically distinct (Bodmin and Boscastle; Migdale $F_{ST}$ values around 0.06, Bodmin and Boscastle; Belfast $F_{ST}$ values around 0.045). Belfast was more genetically different from Migdale than from the populations in the far south-west (Bodmin and Boscastle; Belfast around 0.045, Migdale; Belfast 0.054). Indeed, Migdale was the only population where pairwise $F_{ST}$ exceeded 0.05 which is considered moderate genetic differentiation [26]. This pattern was also reflected in the values for Nei's genetic distance between populations and the principal component analysis, which clearly shows the multilocus genotypes for Migdale form a distinct cluster.

Taken together, these results suggest that the Migdale population in Scotland is isolated, having split off from another population with little gene flow arising after the split, perhaps resulting in a founder effect. That the genetic difference between the Migdale population and the other populations in southern mainland Britain is greater than the difference between the Belfast population and the two southern populations, suggests that a barrier to gene flow other than the Irish Sea exists. This could be a physical barrier, like the Scottish Highlands, or it

could be due to local adaptation or phenological differences. While the climate is rather different in Migdale compared with the other sites, which could lead to differences in phenology that act as a barrier to mating, our observations of multiple populations in the UK suggest that a phenological barrier is unlikely. Under social nesting, the second-generation (B2) males usually emerge in late July to early August. In Migdale where nesting is solitary, we have seen first generation (B1) males from late July until September. Moreover, a recent study by Gruber and Field [12] estimated the maximum survival for a male *H. rubicundus* to be 22 days. As such, it seems unlikely that a lack of emergence synchronicity could consistently preclude matings between males and females from populations with different social structures and phenologies. Another non-mutually exclusive possibility is that these bees are locally adapted to the climate in Migdale and outbreeding with non-locally adapted individuals may result in low fitness offspring which do not survive and reproduce, removing their alleles from the population. There is some evidence for local adaptation to the climate in *H. rubicundus*; bees from the Belfast and Migdale populations in the north have denser sense organs on their antennae that detect humidity, temperature and $CO_2$ than bees from south-western populations in the UK [7]. In future, longitudinal studies using next-generation sequencing to assess allele frequencies across multiple populations of *H. rubicundus* would provide more power to assess the relative importance of local adaptation vs physical barriers as barriers to gene flow in this species. Such studies would also help to elucidate the sex-determining mechanism in this species and predict the risk of a diploid male extinction vortex.

Soro et al. [4] previously found no evidence of genetic differentiation of *H. rubicundus* populations across mainland UK, but this study found that when populations from the extreme north and south are considered, this no longer holds true. *H. rubicundus* collected in Northern Ireland and those collected in southern England were more genetically similar to each other than Scottish bees from Migdale. This suggests that the Irish Sea may not be the only barrier to gene flow. If these bees are locally adapted to the climate, the consequences of climate change for this species may not be as optimistic as predicted by Schürch et al. [8]. In populations with less genetic variation, such as Migdale and Bodmin, environmental stressors resulting from climate change could impose selection which further reduces genetic variation. This could be detrimental to population persistence unless it is countered by increased gene flow with other populations. If the barrier to gene flow between Migdale and other *H. rubicundus* populations is phenological or related to social structure (i.e., is not due to a physical barrier) this may break down under climate change, preserving the population, but our results suggest that this is unlikely. Alternatively, if gene flow is restricted by a physical barrier bees may be more at risk in the far north, where the ecosystem services provided by *H. rubicundus* are most important and pollinator diversity is declining most rapidly [2]. Studies that assess population genetics of *H. rubicundus* at smaller spatial scales within Scotland and the north of England would help to uncover the nature of any barriers to gene flow, providing more accurate assessments of isolation-by-distance so that we can better understand the issues that this important pollinator is likely to face in the present climate crisis. Including populations from the central belt of Scotland would be particularly worthwhile as this area was not included in this or the previous study by Soro et al [4].

The results of this study demonstrate the importance of sampling populations at the limits of a species range for comprehensively assessing patterns of geneflow. We show that the Irish sea is not the only barrier to gene flow in UK *H. rubicundus* populations. By including populations at extreme latitudes in our study, we show that phenology and the climate may act alongside physical barriers including the Irish Sea, and also the Scottish Highlands, to restrict gene flow in *H. rubicundus*.

## Supporting information

**S1 Table. Details of the primers used in each of the 3 sets.**
(DOCX)

**S1 File. Raw data files and code to reproduce analyses can be found in the following data repository: https://osf.io/s8txc/?view_only=aebe7992b4cb4285a90a0e6eb2603685.**
(DOCX)

**S2 File. Extended methods.**
(DOCX)

## Acknowledgments

Many thanks to Stephen Quinn (community park manager, Belfast city council), Ross Watson (Woodland trust, Migdale and Ledmore), Jeff Cherrington (lead ranger, Boscastle to Morwenstow, National Trust), John Keast (Bodmin commons council), Robert Clark (landowner at Bodmin), Martin Wright (common land and village greens registration officer, natural environment and open spaces service, Cornwall council), Lynne Jones (St. Neot Parish council) and South West Water for facilitating access to field sites.

## Author Contributions

**Conceptualization:** Jeremy Field, Bart A. Pannebakker, Rebecca A. Boulton.

**Data curation:** Bas A. Michels, Jodie Gruber, Charlotte Savill, Rebecca A. Boulton.

**Formal analysis:** Bas A. Michels, Mariska M. Beekman, Bart A. Pannebakker.

**Investigation:** Bas A. Michels, Rebecca A. Boulton.

**Methodology:** Bas A. Michels, Mariska M. Beekman.

**Project administration:** Jodie Gruber, Charlotte Savill.

**Resources:** Jodie Gruber, Charlotte Savill.

**Supervision:** Mariska M. Beekman, Bart A. Pannebakker, Rebecca A. Boulton.

**Visualization:** Bas A. Michels.

**Writing – original draft:** Bas A. Michels, Rebecca A. Boulton.

**Writing – review & editing:** Bas A. Michels, Jeremy Field, Jodie Gruber, Bart A. Pannebakker, Charlotte Savill, Rebecca A. Boulton.

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
