## [Decision Letter · Decision Letter 0]

7 Nov 2023

PONE-D-23-27285Genetic differentiation at extreme latitudes in the socially plastic sweat bee Halictus rubicundusPLOS ONE

Dear Dr. Boulton,

Thank you for submitting your manuscript to PLOS ONE. After careful consideration, we feel that it has merit but does not fully meet PLOS ONE’s publication criteria as it currently stands. Therefore, we invite you to submit a revised version of the manuscript that addresses the points raised during the review process.

ACADEMIC EDITOR: The manuscript requires comprehensive revision for the improvement of the draft. Some suggested changes are in the comments portion of the reviewers to revise and improve the manuscript. Please find suggested corrections, reference writing, journal-style format, author’s instructions, use of abbreviations and missing information for revision. Please submit your revised manuscript by Dec 22 2023 11:59PM. If you will need more time than this to complete your revisions, please reply to this message or contact the journal office at plosone@plos.org. Please include the following items when submitting your revised manuscript:A rebuttal letter that responds to each point raised by the academic editor and reviewer(s). You should upload this letter as a separate file labeled 'Response to Reviewers'.A marked-up copy of your manuscript that highlights changes made to the original version. You should upload this as a separate file labeled 'Revised Manuscript with Track Changes'.An unmarked version of your revised paper without tracked changes. You should upload this as a separate file labeled 'Manuscript'.

We look forward to receiving your revised manuscript.

Kind regards,

Bilal Rasool, PhD

Academic Editor

PLOS ONE

Journal Requirements:

"This work is part of a project that received funding from the European Research Council (ERC) under the European Horizon's 202 research and innovation programme (grantagreement no. 695744). RAB was funded by a Wageningen Graduate School Postdoctoral Talent fellowship and a BBSRC discovery fellowship."

"Many thanks to Stephen Quinn (community park manager, Belfast city council), Ross Watson (Woodland trust, Migdale and Ledmore), Jeff Cherrington (lead ranger, Boscastle to Morwenstow, National Trust), John Keast (Bodmin commons council), Robert Clark (landowner at Bodmin), Martin Wright (common land and village greens registration officer, natural environment and open spaces service, Cornwall council), Lynne Jones (St. Neot Parish council) and South West Water for facilitating access to field sites. This work is part of a project that received funding from the European Research Council (ERC) under the European Horizon's 2020 research and innovation programme (grantagreement no. 695744). RB was funded by a Wageningen Graduate School Postdoctoral Talent fellowship and a BBSRC discovery fellowship."

"This work is part of a project that received funding from the European Research Council (ERC) under the European Horizon's 202 research and innovation programme (grantagreement no. 695744). RAB was funded by a Wageningen Graduate School Postdoctoral Talent fellowship and a BBSRC discovery fellowship."

7. We note that you have referenced (Boulton, unpublished data) on page 13, which has currently not yet been accepted for publication. Please remove this from your References and amend this to state in the body of your manuscript: (ie “Bewick et al. [Unpublished]”) as detailed online in our guide for authors

8. Please include your full ethics statement in the ‘Methods’ section of your manuscript file. In your statement, please include the full name of the IRB or ethics committee who approved or waived your study, as well as whether or not you obtained informed written or verbal consent. If consent was waived for your study, please include this information in your statement as well. 

9. We note that [Figure 2] in your submission contain [map/satellite] images which may be copyrighted. All PLOS content is published under the Creative Commons Attribution License (CC BY 4.0), which means that the manuscript, images, and Supporting Information files will be freely available online, and any third party is permitted to access, download, copy, distribute, and use these materials in any way, even commercially, with proper attribution. For these reasons, we cannot publish previously copyrighted maps or satellite images created using proprietary data, such as Google software (Google Maps, Street View, and Earth). For more information, see our copyright guidelines: http://journals.plos.org/plosone/s/licenses-and-copyright.

Additional Editor Comments: 

The manuscript requires comprehensive revision for the improvement of the draft. 

Reviewers' comments:

Reviewer's Responses to Questions

**Comments to the Author**

1. Is the manuscript technically sound, and do the data support the conclusions?

Reviewer #1: Yes

Reviewer #2: Partly

Reviewer #3: Yes

2. Has the statistical analysis been performed appropriately and rigorously? 

Reviewer #1: Yes

Reviewer #2: Yes

Reviewer #3: Yes

3. Have the authors made all data underlying the findings in their manuscript fully available?

Reviewer #1: Yes

Reviewer #2: No

Reviewer #3: No

4. Is the manuscript presented in an intelligible fashion and written in standard English?

Reviewer #1: Yes

Reviewer #2: Yes

Reviewer #3: Yes

5. Review Comments to the Author

Reviewer #1: The paper presents the results of a study aiming to understand whether differences in phenology due to

environmental conditions could limit gene flow in Halictus rubicundus bees. My impression is that the

data are interesting and, with a few adjustments, would make a nice paper in PLOS ONE. Follow my

comments:

1) In general, in the introduction, I think it misses a description of the types of barriers, in addition to

the distance between populations, that could influence bee gene flow. The authors cite the Irish Sea and

temperature differences, but between the south and north of the UK, there are possibly other barriers

that could be better described. My knowledge about UK geography is poor, just like many readers

abroad. So, the detail of this point is important to strengthen the work.

2) Lines 43–44: “Halictus rubicundus (Hymenoptera: Halictidae) is a locally abundant species with a

broad geographic range”. Where are the bees locally abundant? In the north of the globe? In UK? I

understand your point, but starting the text with this information without giving a basis to understand it

doesn’t seem right.

3) Lines 205-207: “Migdale was the most genetically different from all other populations (Bodmin:

FST = 0.061; Nei’s = 0.618. Boscastle: FST = 0.059; Nei’s = 0.625. Belfast: FST = 0.054; Nei’s =

0.491)”. I suggest adding the values of Migdale at this point in the text to facilitate the comparison.

4) Lines 207-210: “Bees collected in the south of mainland Britain (Bodmin and Boscastle) were more

genetically similar to bees across the Irish Sea (Belfast) than they were to bees collected in the North of

mainland Britain in Migdale (Bodmin: FST = 0.046; Nei’s = 0.504. Boscastle: F ST = 0.044; Nei’s =

0.503).” Again, please add the values of all points cited in the text.

Reviewer #2: Article number PONE-D-23-27285 “Genetic differentiation at extreme latitudes in the socially plastic sweat bee Halictus rubicundus” carried results that need revision for the improvement of the draft. Moreover, the conclusion portion should be further developed based on the findings and future implications of the study. Some suggested changes are in the comments portion to revise and improve the manuscript. Please find suggested corrections, Reference writing, journal-style format, author’s instructions, use of abbreviations and missing information for revision.

Abstract: Abstract is lengthy, please be precise in the abstract writings.

Line 45: Please recheck the abbreviations H. rubicundus” and write complete scientific name at the start of the sentence.

Line 45-48: Please rephrase the sentence

Line 48: (the number of generations per year)” remove the sentence

Line 53-54: (a single generation per 54 year)” please remove the sentence.

Line 54: see also Hogendoorn and Leys 1997” please follow the author’s instructions and journal style format while writing the references” this may be recheck throughout the manuscript

Figure 1 may be placed as supplementary figure 1 (figure taken from Schürch et al. 2016) while mentioning the Figure S1 in the text.

Material and methods

The authors collected samples in 2019-2020 and conducted analysis? There is no data found regarding the background of the samples and even the generational data?

Line 160: HWE?

Line 249: to the” remove from the sentence

Please add a precise conclusion

Please write the references in the text and in the reference portion according to the author's instructions and Journal style/formatting. Please double-check for typos and inconsistencies in Journal style/formatting/ authors instructions, double spaces, spellings of the words, English vocabulary, missing italics, scientific names, missing information

Reviewer #3: This article presents a study on the genetic differentiation seen in different populations of Halictus rubicundus in the UK. The paper addresses an interesting topic and is well written; I don’t have comments about the analyses or the interpretation of the results. I feel the discussion could be fleshed out a bit, however, and so I recommend it for publication in PLOS One pending the minor revisions to the discussion outlined below.

The results of the study uncover genetic differentiation between populations in the north and the south of the UK and the authors state that their results “improve understanding of barriers to gene flow”. While the authors suggest that this differentiation may be caused by both phenological and /or physical barriers to gene flow, the study does not appear to have made any attempt to separate these factors. To truly understand the barriers to gene flow, and thus the potential consequences of climate change or habitat degradation on these populations, separating phenological barriers from physical barriers seems essential. Perhaps it was not possible in the present study…but how could this be done in a future study? To highlight geographical barriers - would it be useful to increase the sampled nests to include populations from along the north-south gradient of the UK? If the Scottish Highlands truly represent a barrier to gene flow, then populations in the Scottish lowlands should also be quite different from the Migdale populations. The Irish Sea was not found to have been a total barrier to gene flow (Belfast populations overlapped quite a bit with the Cornish and, to a lesser degree, with the Migdale). Maybe this should be discussed in more detail, given what we know about dispersal capacities of species of bees across bodies of water? Or maybe there’s another explanation for this overlap, i.e. similar alleles that have been retained in different populations despite reproductive isolation? And to look at phenological barriers to gene flow - could one take into consideration voltinism when comparing genetic differentiation (i.e. is there more gene flow between populations that are either univoltine or bivoltine, and less among populations with different social structure?)?

Finally, to the question "Have the authors made all data underlying the findings in their manuscript fully available?", I answered no because I couldn't find any mention of where the microsat markers generated in the study were stored...

6. PLOS authors have the option to publish the peer review history of their article (what does this mean?). If published, this will include your full peer review and any attached files.

Reviewer #1: No

Reviewer #2: No

Reviewer #3: **Yes: **Jesse Litman

---

## [Author Response · Author response to Decision Letter 0]

15 Jan 2024

Reviewer #1: The paper presents the results of a study aiming to understand whether differences in phenology due to environmental conditions could limit gene flow in Halictus rubicundus bees. My impression is that the data are interesting and, with a few adjustments, would make a nice paper in PLOS ONE. Follow my comments: 

We thank reviewer #1 for these kind words. 

1) In general, in the introduction, I think it misses a description of the types of barriers, in addition to the distance between populations, that could influence bee gene flow. The authors cite the Irish Sea and temperature differences, but between the south and north of the UK, there are possibly other barriers that could be better described. My knowledge about UK geography is poor, just like many readers abroad. So, the detail of this point is important to strengthen the work. 

We agree that this part of the introduction can be extended, so we have also introduced the Scottish Highlands as a possible barrier between the south and north, and some other barriers (L97-103). 

2) Lines 43–44: “Halictus rubicundus (Hymenoptera: Halictidae) is a locally abundant species with a broad geographic range”. Where are the bees locally abundant? In the north of the globe? In UK? I understand your point, but starting the text with this information without giving a basis to understand it doesn’t seem right. 

We have changed this sentence to be more specific (L38-40). 

3) Lines 205-207: “Migdale was the most genetically different from all other populations (Bodmin: FST = 0.061; Nei’s = 0.618. Boscastle: FST = 0.059; Nei’s = 0.625. Belfast: FST = 0.054; Nei’s = 0.491)”. I suggest adding the values of Migdale at this point in the text to facilitate the comparison. 

These values shown here are values comparing Migdale with Bodmin, Boscastle and Belfast. We agree see that this might not be clear from the way it was expressed, and have added a table (Table 4) displaying these values so there may be no misunderstanding of how to interpret the results.

4) Lines 207-210: “Bees collected in the south of mainland Britain (Bodmin and Boscastle) were more genetically similar to bees across the Irish Sea (Belfast) than they were to bees collected in the North of mainland Britain in Migdale (Bodmin: FST = 0.046; Nei’s = 0.504. Boscastle: F ST = 0.044; Nei’s = 0.503).” Again, please add the values of all points cited in the text. 

This is the same issue as with comment #3, and we think the addition of the Table 4 will also be beneficial in the presentation of the data for these comparisons. 

Reviewer #2: Article number PONE-D-23-27285 “Genetic differentiation at extreme latitudes in the socially plastic sweat bee Halictus rubicundus” carried results that need revision for the improvement of the draft. Moreover, the conclusion portion should be further developed based on the findings and future implications of the study. Some suggested changes are in the comments portion to revise and improve the manuscript. Please find suggested corrections, Reference writing, journal-style format, author’s instructions, use of abbreviations and missing information for revision. 

We thank the reviewer for their feedback on our study and have adjusted the manuscript, based on their suggestions, as described below.

Abstract: Abstract is lengthy, please be precise in the abstract writings. 

We have reduced the abstract significantly for conciseness. 

Line 45: Please recheck the abbreviations H. rubicundus” and write complete scientific name at the start of the sentence. 

We have written out the complete scientific name at the start of the sentence (L41).

Line 45-48: Please rephrase the sentence 

We are unsure what the reviewer would like us to change about this sentence. We are happy to edit with further comments from the reviewer. 

Line 48: (the number of generations per year)” remove the sentence 

This sentence has been removed (L43).

Line 53-54: (a single generation per 54 year)” please remove the sentence. 

This sentence has been removed (L47).

Line 54: see also Hogendoorn and Leys 1997” please follow the author’s instructions and journal style format while writing the references” this may be recheck throughout the manuscript 

https://www.researchgate.net/post/How_to_auto_convert_references_from_word_file_to_Mendeley

We thank the reviewer for pointing this out. We have changed the reference format accordingly. 

Figure 1 may be placed as supplementary figure 1 (figure taken from Schürch et al. 2016) while mentioning the Figure S1 in the text. 

We understand the point reviewer #2 is making here. However, we feel that Figure 1 is essential to the introduction, as a big part of this paper is about the life cycle of H. rubicundus and how differences could influence phenology and gene flow. Therefore, we believe that a Figure 1 is an important visual aid to set up the study and so we have left it as a main figure. We are happy to be guided by the editors preference with regards to moving Figure 1 to the supplementary material. 

Material and methods 

The authors collected samples in 2019-2020 and conducted analysis? There is no data found regarding the background of the samples and even the generational data? 

We thank the reviewer for raising this point. We have added this information to the data repository, for which the link can be found in the supplementary section. This is the link to the meta-data with details about collection of the samples https://osf.io/pvj56?view_only=aebe7992b4cb4285a90a0e6eb2603685

Line 160: HWE? 

We have added ‘Hardy-Weinberg equilibrium’ in full (L159).

Line 249: to the” remove from the sentence 

Done.

Please add a precise conclusion 

We thank reviewer #2 for their feedback regarding our conclusion. We have added a more precise conclusion (L388-393).

Please write the references in the text and in the reference portion according to the author's instructions and Journal style/formatting. Please double-check for typos and inconsistencies in Journal style/formatting/ authors instructions, double spaces, spellings of the words, English vocabulary, missing italics, scientific names, missing information 

Done.

Reviewer #3: This article presents a study on the genetic differentiation seen in different populations of Halictus rubicundus in the UK. The paper addresses an interesting topic and is well written; I don’t have comments about the analyses or the interpretation of the results. I feel the discussion could be fleshed out a bit, however, and so I recommend it for publication in PLOS One pending the minor revisions to the discussion outlined below.

We thank reviewer #3 for their kind words, and have fleshed out the discussion, specifically on future research recommendations, as per their suggestion. 

The results of the study uncover genetic differentiation between populations in the north and the south of the UK and the authors state that their results “improve understanding of barriers to gene flow”. While the authors suggest that this differentiation may be caused by both phenological and /or physical barriers to gene flow, the study does not appear to have made any attempt to separate these factors. To truly understand the barriers to gene flow, and thus the potential consequences of climate change or habitat degradation on these populations, separating phenological barriers from physical barriers seems essential. Perhaps it was not possible in the present study…but how could this be done in a future study? To highlight geographical barriers - would it be useful to increase the sampled nests to include populations from along the north-south gradient of the UK? If the Scottish Highlands truly represent a barrier to gene flow, then populations in the Scottish lowlands should also be quite different from the Migdale populations. The Irish Sea was not found to have been a total barrier to gene flow (Belfast populations overlapped quite a bit with the Cornish and, to a lesser degree, with the Migdale). Maybe this should be discussed in more detail, given what we know about dispersal capacities of species of bees across bodies of water? Or maybe there’s another explanation for this overlap, i.e. similar alleles that have been retained in different populations despite reproductive isolation? And to look at phenological barriers to gene flow - could one take into consideration voltinism when comparing genetic differentiation (i.e. is there more gene flow between populations that are either univoltine or bivoltine, and less among populations with different social structure?)

We agree with the suggestion of reviewer #3 to include populations from the Scottish lowlands (the central belt) in future comparisons, to test whether the Scottish highlands are indeed the main barrier to gene flow (L361-363).

Regarding the comment on the Irish Sea from reviewer #3, we still mention that the Irish Sea is a barrier to gene flow, just not as much as expected (L249-252). We think that it is very unlikely that the similarity is due to similar alleles being retained over time, as we are working with microsatellites here, which should be selectively neutral. 

Finally, to the question "Have the authors made all data underlying the findings in their manuscript fully available?", I answered no because I couldn't find any mention of where the microsat markers generated in the study were stored... 

We have added the raw unscored data files (.fsa format) which can now be accessed via the repository here https://osf.io/s8txc/?view_only=aebe7992b4cb4285a90a0e6eb2603685, as also described in the supporting information in the manuscript (L38

---

## [Decision Letter · Decision Letter 1]

11 Mar 2024

PONE-D-23-27285R1Genetic differentiation at extreme latitudes in the socially plastic sweat bee Halictus rubicundusPLOS ONE

Dear Dr. Boulton,

Thank you for submitting your manuscript to PLOS ONE. After careful consideration, we feel that it has merit but does not fully meet PLOS ONE’s publication criteria as it currently stands. Therefore, we invite you to submit a revised version of the manuscript that addresses the points raised during the review process.

**ACADEMIC EDITOR: Please revise the draft according to comments of the reviewers ** 

Kind regards,

Bilal Rasool, PhD

Academic Editor

PLOS ONE

Journal Requirements:

Reviewers' comments:

Reviewer's Responses to Questions

**Comments to the Author**

1. If the authors have adequately addressed your comments raised in a previous round of review and you feel that this manuscript is now acceptable for publication, you may indicate that here to bypass the “Comments to the Author” section, enter your conflict of interest statement in the “Confidential to Editor” section, and submit your "Accept" recommendation.

Reviewer #1: All comments have been addressed

Reviewer #2: (No Response)

Reviewer #3: All comments have been addressed

2. Is the manuscript technically sound, and do the data support the conclusions?

Reviewer #1: Yes

Reviewer #2: Yes

Reviewer #3: Yes

3. Has the statistical analysis been performed appropriately and rigorously? 

Reviewer #1: Yes

Reviewer #2: Yes

Reviewer #3: Yes

4. Have the authors made all data underlying the findings in their manuscript fully available?

Reviewer #1: Yes

Reviewer #2: Yes

Reviewer #3: Yes

5. Is the manuscript presented in an intelligible fashion and written in standard English?

Reviewer #1: Yes

Reviewer #2: Yes

Reviewer #3: Yes

6. Review Comments to the Author

Reviewer #1: I found the revised manuscript much easier to read than the previous version. Particularly, the results seems clear to me now. I think this is an interesting study worth publishing.

Reviewer #2: The revised article, PONE-D-23-27285R1, "Genetic differentiation at extreme latitudes in the socially plastic sweat bee Halictus rubicundus" has improved after revision; however, some issues still need to be verified before the draft is officially considered.

The authors feel that “figure 1 is essential to the introduction, as a big part of this paper is about the life cycle of H. rubicundus and how differences could influence phenology and gene flow. Therefore, we believe that figure 1 is an important visual aid to set up the study and so we have left it as a main figure.......” Kindly recheck and consult for the journal's policies with the managing editor. "It is okay with me if the journal policies permit the reproduction of figure 1 from the earlier literature.

The authors should also recheck the formatting guidelines for tables. Journal-style formatting and adjustments should be made as needed.

Line 380: Reference [2] please recheck if this reference is relevant to the text (if necessary)

Could you please confirm that the references in the discussion section follow the numerical sequence, as they appear to be out of order? (Make changes if necessary.)

Please write the references in the text and in the reference portion according to the author's instructions and Journal's style and formatting. Please double-check for typos and inconsistencies in Journal style formatting, author's instructions, double spaces, spellings of the words, English vocabulary, missing italics, scientific names, and missing information etc.

Reviewer #3: The authors have addressed the comments from three different reviewers and I feel the manuscript is now ready for publication in PLOS One.

7. PLOS authors have the option to publish the peer review history of their article (what does this mean?). If published, this will include your full peer review and any attached files.

Reviewer #1: No

Reviewer #2: No

Reviewer #3: No

---

## [Author Response · Author response to Decision Letter 1]

24 Mar 2024

Reviewer #2: The revised article, PONE-D-23-27285R1, "Genetic differentiation at extreme latitudes in the socially plastic sweat bee Halictus rubicundus" has improved after revision; however, some issues still need to be verified before the draft is officially considered.

The authors feel that “figure 1 is essential to the introduction, as a big part of this paper is about the life cycle of H. rubicundus and how differences could influence phenology and gene flow. Therefore, we believe that figure 1 is an important visual aid to set up the study and so we have left it as a main figure.......” Kindly recheck and consult for the journal's policies with the managing editor. "It is okay with me if the journal policies permit the reproduction of figure 1 from the earlier literature.

We have made our own version of the figure from Shurch et al, and this is now used as figure 1. We still credit the original paper in the legend due to the similar format. 

The authors should also recheck the formatting guidelines for tables. Journal-style formatting and adjustments should be made as needed.

Done

Line 380: Reference [2] please recheck if this reference is relevant to the text (if necessary)

We have kept this reference in (now line 358 in the track changed version of the ms) this statement refers to the greater declines in pollinators in the north and this reference (where we originally read this) is the most appropriate. 

Could you please confirm that the references in the discussion section follow the numerical sequence, as they appear to be out of order? (Make changes if necessary.)

Done

Please write the references in the text and in the reference portion according to the author's instructions and Journal's style and formatting. Please double-check for typos and inconsistencies in Journal style formatting, author's instructions, double spaces, spellings of the words, English vocabulary, missing italics, scientific names, and missing information etc.

Done

---

## [Decision Letter · Decision Letter 2]

10 Apr 2024

Genetic differentiation at extreme latitudes in the socially plastic sweat bee Halictus rubicundus

PONE-D-23-27285R2

Dear Dr. Rebecca,

We’re pleased to inform you that your manuscript has been judged scientifically suitable for publication and will be formally accepted for publication once it meets all outstanding technical requirements.

Kind regards,

Bilal Rasool, PhD

Academic Editor

PLOS ONE

Reviewers' comments:

Reviewer's Responses to Questions

**Comments to the Author**

1. If the authors have adequately addressed your comments raised in a previous round of review and you feel that this manuscript is now acceptable for publication, you may indicate that here to bypass the “Comments to the Author” section, enter your conflict of interest statement in the “Confidential to Editor” section, and submit your "Accept" recommendation.

Reviewer #2: All comments have been addressed

2. Is the manuscript technically sound, and do the data support the conclusions?

Reviewer #2: Partly

3. Has the statistical analysis been performed appropriately and rigorously? 

Reviewer #2: Yes

4. Have the authors made all data underlying the findings in their manuscript fully available?

Reviewer #2: Yes

5. Is the manuscript presented in an intelligible fashion and written in standard English?

Reviewer #2: Yes

6. Review Comments to the Author

Reviewer #2: The article is much improved after revisions, however, formatting of figures and tables according to journal style may be considered in proofreading.

7. PLOS authors have the option to publish the peer review history of their article (what does this mean?). If published, this will include your full peer review and any attached files.

Reviewer #2: No

---

## [Editor Report · Acceptance letter]

26 Apr 2024

PONE-D-23-27285R2 

PLOS ONE

Dear Dr. Boulton, 

I'm pleased to inform you that your manuscript has been deemed suitable for publication in PLOS ONE. Congratulations! Your manuscript is now being handed over to our production team.

Kind regards, 

on behalf of

Dr Bilal Rasool 

Academic Editor

PLOS ONE